# Gout and Colorectal Cancer Likelihood: Insights from a Nested Case-Control Study of the Korean Population Utilizing the Korean National Health Insurance Service-National Sample Cohort

**DOI:** 10.3390/cancers15235602

**Published:** 2023-11-27

**Authors:** Mi Jung Kwon, Kyeong Min Han, Joo-Hee Kim, Ji Hee Kim, Min-Jeong Kim, Nan Young Kim, Hyo Geun Choi, Ho Suk Kang

**Affiliations:** 1Department of Pathology, Hallym University Sacred Heart Hospital, Hallym University College of Medicine, Anyang 14068, Republic of Korea; mulank@hanmail.net; 2Hallym Data Science Laboratory, Hallym University College of Medicine, Anyang 14068, Republic of Korea; 3Division of Pulmonary, Allergy, and Critical Care Medicine, Department of Medicine, Hallym University Sacred Heart Hospital, Hallym University College of Medicine, Anyang 14068, Republic of Korea; luxjhee@gmail.com; 4Department of Neurosurgery, Hallym University Sacred Heart Hospital, Hallym University College of Medicine, Anyang 14068, Republic of Korea; kimjihee.ns@gmail.com; 5Department of Radiology, Hallym University Sacred Heart Hospital, Hallym University College of Medicine, Anyang 14068, Republic of Korea; drkmj@hallym.or.kr; 6Hallym Institute of Translational Genomics and Bioinformatics, Hallym University Medical Center, Anyang 14068, Republic of Korea; 7Suseo Seoul E.N.T. Clinic and MD Analytics, Seoul 06349, Republic of Korea; mdanalytics@naver.com; 8Department of Internal Medicine, Hallym University Sacred Heart Hospital, Hallym University College of Medicine, Anyang 14068, Republic of Korea

**Keywords:** colorectal cancer, gout, nested case-control study, national healthcare data

## Abstract

**Simple Summary:**

Using Korean National Health Insurance Service-National Sample Cohort database, we studied whether patient-specific factors such as age, gender, socioeconomic status, and presence of comorbidities may influence the relationship between gout and the likelihood of developing CRC. Our nested case-control study with propensity score overlap-weighted multivariate logistic regression analyses suggest that patients with gout are not at higher risk for CRC than a control group, indicating that gout may not significantly and independently contribute to the CRC risk of the general population. Nevertheless, within a specific subgroup of patients aged less than 65 years, there appeared to be a moderately reduced probability of CRC development. Further comprehensive research is necessary to substantiate this association and gain a better understanding of the underlying causal factors.

**Abstract:**

Considering the global importance of both gout and colorectal cancer (CRC) as significant health issues with mutual relevance, we aimed to examine the risk of colorectal cancer in Korean patients with gout. In this nested case-control study, we used data from 9920 CRC patients and 39,680 controls the Korean National Health Insurance Service-National Sample Cohort database. Propensity score overlap-weighted multivariate logistic regression analyses, adjusted for confounders, were used to assess the odds ratio (OR) and 95% confidence interval (CI) of the association between gout and CRC. Adjusted OR for CRC were similar between patients with gout and the control group (0.95; 95% CI, 0.86–1.04; *p* = 0.282). However, after adjustment, subgroup analysis revealed an 18% reduction in the probability of CRC among patients younger than 65 years with gout (95% CI, 0.70–0.95; *p* = 0.009). Conversely, absence of an association between gout and subsequent CRC persisted regardless of sex, income, residence, and Charlson Comorbidity Index score, even among individuals aged 65 years or older. These results imply that gout may not be a significant independent risk factor for CRC among the general population. However, in patients younger than 65 years with gout, a slightly reduced likelihood of CRC was observed. Further research is necessary to establish a causal relationship between gout and CRC and to generalize these findings to other populations.

## 1. Introduction

Gout is the most widespread form of inflammatory arthritis and a substantial contributor to disability according to the Global Burden of Diseases Study, and it is increasingly becoming a global health concern. Its prevalence and incidence have alarmingly doubled over the past three decades. This systemic and metabolic disorder is strongly associated with elevated uric acid levels and progressive accumulation of monosodium urate crystals in the joints and throughout the body over time [1]. Gout is also associated with various systemic comorbidities, including metabolic syndrome, obesity, chronic kidney disease, cardiovascular diseases [1,2], and malignancy [3].

Colorectal cancer (CRC) is the third most common cancer and the second leading cause of cancer-related deaths worldwide, accounting for 10% of all cancer cases and fatalities [4]. In 2018, South Korea had the world’s second highest incidence of CRC, with 44.5 cases per 100,000 individuals annually, leading to a significant increase in its disease burden and making CRC the country’s third most common cancer [5]. CRC can serve as an indicator of socioeconomic development because it is influenced by factors such as aging, urbanization, and changes in lifestyle and dietary habits [6], all of which are independently associated with an increased risk of CRC [7,8]. These changes have occurred rapidly in Korea over the past 10 to 20 years [7,8]. Therefore, there is an impending need to identify primary modifiable risk factors responsible for the increased incidence of CRC.

A growing body of research has shown that gout can increase the risk of cancer [3,9,10,11,12,13]. It has been suggested that the dual pro-oxidant and antioxidant properties of uric acid may play an important role in the process of carcinogenesis [14]. The importance of the connection between chronic systemic inflammation and this relationship is further emphasized by a substantial body of evidence that links persistent inflammation to various gastrointestinal cancers, including CRC [15]. However, reports on the association between gout and CRC are conflicting. Some studies have reported a significant positive association between gout and CRC [10,11,13], whereas others have not [3,12,16]. One cross-sectional study suggested a potential protective effect of gout against CRC compared to other inflammatory arthritides [17]. However, many of these studies had imbalanced sample sizes and included cohorts with non-uniform demographic data, such as sex, age, or socioeconomic level [10,11,12,13,16,17], which may have introduced a selection bias. Additionally, the methods used to select control subjects were often unclear [3,11], lacked matching [10,17], or involved limited matching for age and sex [12,13,16] between comparison groups. Two epidemiological studies examined the likelihood of malignancy development in patients with gout in Korea [12,13]. However, there is a lack of consensus on this relationship and the role of factors such as age, sex, socioeconomic status, or comorbidities influencing the risk of CRC in patients with gout [12,13]. Moreover, estimates of the risk of CRC in Korean patients with gout from these studies have been varied. Three studies [3,10,11] identified a positive association between gout and digestive system cancers, including CRC. However, significant variations among studies have made it challenging to establish a specific link between gout and CRC development [9]. Therefore, further validation using national population cohort data with well-balanced demographics is essential to mitigate the influence of confounding factors.

In this study, our objective was to investigate the risk of CRC among Korean individuals with gout. We conducted a meticulously designed nested case-control study, utilizing comprehensive subgroup analyses based on data from the Korean national public healthcare system, in order to explore the potential impact of gout on the development of CRC.

## 2. Materials and Methods

This study was approved by the Ethics Committee of Hallym University, Anyang, South Korea (2022-10-008) and adhered to the guidelines and regulations set forth by this committee. The requirement for written informed consent was waived by the Institutional Review Board.

The data utilized during this study were sourced from the Korean National Health Insurance Service-National Sample Cohort (KNHIS-NSC) [18]. This database consists of 1,137,861 participants and 219,673,817 medical billing codes recorded between January 2002 and December 2019. KNHIS-NSC uses a systematic sampling method to represent a prototypical sample of 1,025,340 people, equivalent to approximately 2.2% of the Korean population in 2002. These individuals were followed for 17 years (until 2019). Additional information on the representativeness of the data and cohort can be found in other relevant publications [19]. 

A nested case-control study design was considered appropriate to examine the relationship between subjects’ medical histories, presence or absence of exposure, and outcomes. The KNHIS-NSC database follows the International Classification of Diseases, 10th revision (ICD-10), codes for the standardization of disease diagnoses and organization of healthcare information. During this study, participants newly diagnosed with CRC between 2005 and 2019 (*n* = 9920) were initially selected based on their diagnosis of CRC (ICD-10 codes: C18, C19, C20, D010, D011, and D012) and the presence of a special claims code indicating severe cancer (V193 or V194). A special claims code indicating severe cancer-related illness that allows for payment reductions for such patients has been implemented by the National Healthcare Service since 2005. Individuals not diagnosed with CRC between 2005 and 2019 were included in the control group (*n* = 1,127,941). Individuals in the control group who had been diagnosed with CRC even once (*n* = 3472) were excluded from the analysis. Individuals in the CRC group whose ICD-10 diagnostic code for CRC was assigned without a special claims code for severe illness or cancer or without a medical history were excluded.

Propensity score matching was performed to reduce discrepancies in baseline demographic and clinical characteristics of the CRC and control groups. Through this process, CRC participants were matched with control participants with similar propensity scores based on age, sex, income, and region of residence. For each patient diagnosed with CRC, the index date was determined as the day when both the ICD-10 codes for CRC and the special claims code (V193 or V194) were electronically assigned to the health insurance claims datasets. The index date for the control group was defined as the index date for matched patients with CRC. Consequently, each matched patient and the control group shared the same index dates. Among the control group, 1,084,789 people were excluded after the matching process. Ultimately, 9920 patients with CRC were matched with 39,680 controls in a 1:4 ratio (Figure 1). Given that research indicates minimal improvements in test power beyond a 1:5 matching ratio, and no appreciable gain in statistical power was observed beyond a 1:4 control-to-case ratio in the study [20], we opted for a 1:4 matching approach in our study. We surveyed patients from both groups with a history of gout based on ICD-10 codes before the index date.

### 2.1. Exposure (Gout)

To ensure accuracy of the analysis by eliminating false-positive cases, only participants who had been diagnosed or managed at least twice according to the ICD-10 code (K05.3) at clinics or hospitals were included in the study [21].

### 2.2. Outcome (Colorectal Cancer)

To maintain analytical accuracy and exclude false-positive cases, CRC cases were determined using specific ICD-10 codes (C18, C19, C20, D010, D011, and D012) allocated for the diagnosis, such as malignant neoplasms of the colon (C18), the rectosigmoid junction (C19), and the rectum (C20) and carcinomas in situ of the colon (D010), the rectosigmoid junction (D011), and the rectum (D012). Among these codes, participants with special claims codes for cancer (V193 or V194) were selected. The use of special claims codes (V193 or V194) indicated the presence of severe cancer and confirmed eligibility for reduced healthcare payments.

### 2.3. Covariates

Participants were categorized into 10 age groups with 5-year intervals and further classified into five income categories, from level 1 (lowest income) to level 5 (highest income). Residential areas were classified into 16 counties according to administrative districts, with further classification into either urban or rural areas [22]. Urban areas comprised the 7 largest cities in Korea (Seoul, Incheon, Daejeon, Daegu, Gwangju, Ulsan, and Busan), each with regional populations exceeding one million, while rural areas encompassed other regions with total populations below one million (Chungcheongbuk, Chungcheongnam, Gyeonggi, Gangwon, Gyeongsangbuk, Gyeongsangnam, Jeollabuk, Jeollanam, and Jeju). The Charlson Comorbidity Index (CCI) was used to measure the disease burden of comorbidities, assigning a sum score ranging from 0–29 based on 17 potential comorbidities [23]. However, during this study, cancer was excluded from the CCI score to examine the potential impact of other comorbidities on CRC development. 

### 2.4. Statistical Analysis

To compare baseline characteristics between groups, standardized differences were used. Propensity score overlap weighting was used to increase the effective sample size and ensure balanced covariates. The propensity score, which was calculated through multivariate logistic regression, including all covariates, was used for the overlap weighting approach. Propensity scores of participants with CRC were weighted based on their probability, whereas propensity scores of control participants were weighted based on the probability of 1 minus the propensity score. This weighting, ranging from 0 to 1, aimed to achieve optimal balance and improve the precision of the analyses [17,24,25]. Standardized differences less than 0.20 were defined as satisfactory to achieve balance, and for standardized differences greater than 0.20, logistic regression was performed to adjust for covariates [26].

To analyze the overlap-weighted odds ratios (ORs) and their corresponding 95% confidence intervals (CIs) for a history of gout and CRC development, a propensity score overlap-weighted multivariable logistic regression analysis was performed. Crude (unadjusted) and overlap-weighted models (adjusted for age, sex, income, region of residence, and CCI score) were used for these analyses. Additionally, subgroup analyses were performed according to age, sex, income, region of residence, and CCI score. 

Two-tailed analyses were performed, and significance was defined as *p* < 0.05. SAS version 9.4 (SAS Institute Inc., Cary, NC, USA) was used for the statistical analyses.

## 3. Results

The study included 9920 patients with CRC who were age-matched, sex-matched, income-matched, and residence-matched with a control group comprising 39,680 individuals from the database. Table 1 presents the demographic characteristics of participants at baseline before and after adjustment for redundancy weights for propensity score matching. Prior to adjustment, age, residence, sex, and income had a standardized difference of 0.00, indicating no disparities between the CRC and control groups. Other basic characteristics, such as CCI scores and history of gout before the index date, were not significantly different between the CRC and control groups (standardized differences = 0.09 and 0.01, respectively). 

After implementing overlap weighting adjustments, the standardized differences for all covariates were substantially reduced, resulting in a well-balanced distribution of demographic characteristics in both groups, with each covariate displaying a standardized difference ≤ 0.2.

### 3.1. Relationship between Gout and Colorectal Cancer

Table 2 shows the crude and adjusted ORs for the CRC incidence of patients with gout. In both the crude and overlap weighting adjusted models, the ORs for CRC incidence were not statistically different between the gout and control groups (crude OR, 0.97; 95% CI, 0.86–1.10; *p* = 0.628; adjusted OR, 0.95; 95% CI, 0.86–1.04; *p* = 0.282). 

### 3.2. Subgroup Analysis

We conducted a more detailed exploration of the relationship between gout and CRC by separating patients based on variables such as age, sex, income, region of residence, and CCI score (Table 2). We observed statistically significant adjusted ORs for certain patient subgroups, such as patients with gout younger than 65 years, who exhibited a slightly lower likelihood of CRC development (adjusted OR, 0.82; 95% CI, 0.70–0.95; *p* = 0.009) after adjustment.

Other subgroup analyses showed a lack of association between gout and CRC development regardless of factors such as sex, income level, residential location, and CCI score, even among individuals aged 65 years or older. 

## 4. Discussion

By applying the propensity score overlap-weighted multivariable logistic regression analysis adjusted for comorbidities, demographic characteristics, and socioeconomic variables, this study revealed that, compared with the control group, all participants with gout may not be at risk for CRC. However, variations in the risk association depending on the specific demographic factor were noted. There was a slightly decreased propensity for CRC within the subgroup of patients younger than 65 years with gout. This finding implies that gout may not be a significant independent risk factor for CRC in the general population. Individual patient characteristics, especially age, may be informative among a subset of patients with gout when assessing the CRC risk. Our results may provide relief from concerns regarding CRC as a gout-related comorbidity.

Our findings are in line with the results of two population-based studies in Taiwan that were based on the National Health Insurance database information during different periods [3,16] and reported no significant association between gout and the risk of CRC. Relevant studies on the impact of gout on incident CRC remain relatively sparse; only five published studies [3,10,11,16,17] and one meta-analysis [9] have been identified. Among them, it is remarkable that only one study specifically focused on the CRC likelihood for patients with gout and discovered no definite association between gout and incident CRC after adjusting for sex, age, residence, and comorbidities (1.03; 95% CI, 0.93–1.14) during 13 years of follow-up [16]. This finding is consistent with our results, in which participants with gout exhibited a likelihood of CRC that was comparable to that of the control group (0.95; 95% CI, 0.86–1.04). Similar to the National Health Insurance claims data, the results of this nested case-control study indicated that, overall, patients with gout do not appear to have an increased risk of CRC compared to that of the control group.

Our findings are in contrast to those of previous studies conducted in Sweden and Taiwan that reported a 1.29-fold to 2.25-fold increased risk of CRC associated with gout [10,11]. The Swedish cohort study highlighted an increased CRC risk, specifically for male patients with gout [10]. These studies included a variety of cancer types with limited sample sizes and did not specifically focus on CRC. Previous studies did not achieve an exact balance between the study and control groups in terms of baseline sociodemographic and health characteristics. Heterogeneity due to demographic differences likely causes large differences in the original quality of the research group [27]. We used nationwide population-based controls matched using propensity scores and adjusted using the overlap weighting method to accurately balance baseline characteristics, minimizing potential study heterogeneity and selection bias. Potential confounding factors were also adjusted by performing multivariate conditional logistic regression analysis. The present study revealed that gout may not be associated with an increased risk of CRC, even after adjusting for confounding factors, including sex, age, residence, income status, and CCI score, consistent with previous studies [3,16]. During this nationwide cohort study, we found that gout might not be a significant independent risk factor for CRC in the general population.

Interestingly, we observed a reduction of 18% (95% CI, 0.70–0.95) in the likelihood of CRC development among a specific subgroup of gout patients aged less than 65 years. Our study also revealed that the risk of CRC was not increased for male or female patients with gout or individuals aged 65 years or older. Previous research did not include sub-analyses to explore how factors such as age, sex, socioeconomic status, or comorbidities might influence the risk of CRC in patients with gout [3,11,12,13,16]. A study in Korea noted a declining trend in the association between gout and CRC risk among middle-aged patients (age, 41–55 years); however, this trend was not statistically significant (0.697; 95% CI, 0.469–1.035) [12], thus aligning with our findings to some extent. Our findings are especially significant for Korean patients with gout because CRC is the second most prevalent malignancy and a major cause of death among both sexes in South Korea, with a high incidence among individuals aged 65 years and older [28]. The most significant takeaway from our study was the finding that patients with gout in the Korean population may not be at increased risk for CRC development, regardless of age. 

The precise mechanism underlying the connection between gout and the reduced risk of CRC in a specific subgroup of individuals younger than age 65 years remains unclear. This diminished risk might be associated with the frequent use of medications, such as colchicine, urate-lowering agents, and anti-inflammatory drugs by patients with gout, potentially contributing to a lower risk within this age group [16]. It has been established that non-steroidal anti-inflammatory drug (NSAID) use is linked to a decreased CRC risk in the general population, and it is plausible that this protective effect extends to individuals with gout. Additionally, colchicine, which is a medication used to manage acute gout attacks because of its microtubule-disrupting properties, is recognized for its anti-cancer effects [29,30]; therefore, it is being studied in nanoparticle form as a potential anti-cancer agent for CRC [31]. In Taiwanese patients with gout and a mean age of 44.63 ± 14.84 years, the use of colchicine was associated with a statistically significant 25% lower risk (95% CI, 0.60–0.94) of CRC compared to that associated with non-use, even after adjusting for age [32]. Another commonly used class of drugs for gout management is xanthine oxidase inhibitors, such as allopurinol. These inhibitors are believed to exert anti-cancer effects by reducing uric acid production and mitigating oxidative stress [33]. Allopurinol use for more than 1 year reduces the risk of prostate cancer by 34% to 36% [34]. Moreover, during a study involving 73 subjects with colonic adenomas, allopurinol use for 4 weeks decreased the oxidative activity of colonic adenomatous polyps and adjacent normal tissues [35]. In a population-based propensity-matched case-control study in Taiwan, there was a statistically significant association between allopurinol use and CRC risk (OR, 0.79; 95% CI, 0.69–0.90), with a dose-response relationship between the two [36]. However, a study on United States veterans found that the 10-year prevalence of CRC was significantly lower among patients with gout (0.8%) than among those with osteoarthritis (3.7%) [17]. CRC was less common among subjects with gout, irrespective of NSAID use, suggesting that NSAID use does not explain the differences in CRC rates [17]. Additionally, patients with gout treated with allopurinol were not at higher risk for all cancers, including CRC, compared to non-users [11], implying the involvement of other underlying mechanisms. Because our study did not specifically investigate the use of anti-inflammatory drugs or urate-lowering agents, we could not confirm their potential therapeutic effects on patients with gout. Nevertheless, the absence of an increased CRC risk for individuals with gout may have significant clinical importance and provide psychological relief to affected individuals. 

This study benefits from a large cohort with a substantial sample size, involving 9920 patients and 39,680 controls, meticulously sourced from a nationwide healthcare database. The study employs sophisticated techniques, including propensity score weighting, to ensure a balance across various covariates. This approach effectively mitigates selection bias across critical demographic factors, including age, sex, income, residence, and CCI scores among participants. As a result, the study closely replicates the characteristics typically encountered in randomized clinical trials [37], significantly reducing the potential for study bias and enhancing its robustness. Demographic heterogeneity within the participant pool may have influenced the nature of associations observed between the original characteristics of populations [38].

This study had some limitations. First, because of its retrospective observational design, we could not establish a definitive causal relationship between gout and CRC. Second, because this study targeted the Korean population and used diagnostic codes from Korean health insurance data, generalizability of the findings to other demographic groups is limited. Finally, the KNHIS-NSC database does not provide extensive information on factors such as the severity of gout, CRC stage, histologic characteristics, tumor progression, survival rates, family history, medication usage, dietary habits, and baseline and evolving uric acid levels during cancer development. Limitations include the absence of separate analyses for colon and rectal cancers with detailed staging and location data, as well as the lack of tumor stage and prognostic information. These limitations may hinder our ability to address potential confounding factors that were not measured, thereby limiting our analysis in accounting for these unmeasured variables. An additional limitation of our study is the inability to analyze body mass index and smoking status, which were not included in the analysis.

## 5. Conclusions

Our comprehensive study suggests that gout does not appear to significantly increase the risk of CRC in the overall Korean population. However, we observed variations in this risk association among specific demographic factors. Notably, within a subgroup of patients aged under 65 years, there was a moderately reduced probability of CRC development among those with gout. Conversely, no such associations were evident in subgroup analyses based on sex, income, residential location, or CCI scores, even among individuals aged 65 years or older. These findings highlight the importance of considering individual patient characteristics, particularly age, when evaluating CRC risk in patients with gout. While our study contributes valuable insights, further research is warranted to explore these associations in greater depth and explore potential mechanisms underlying these observations.

## Figures and Tables

**Figure 1 cancers-15-05602-f001:**
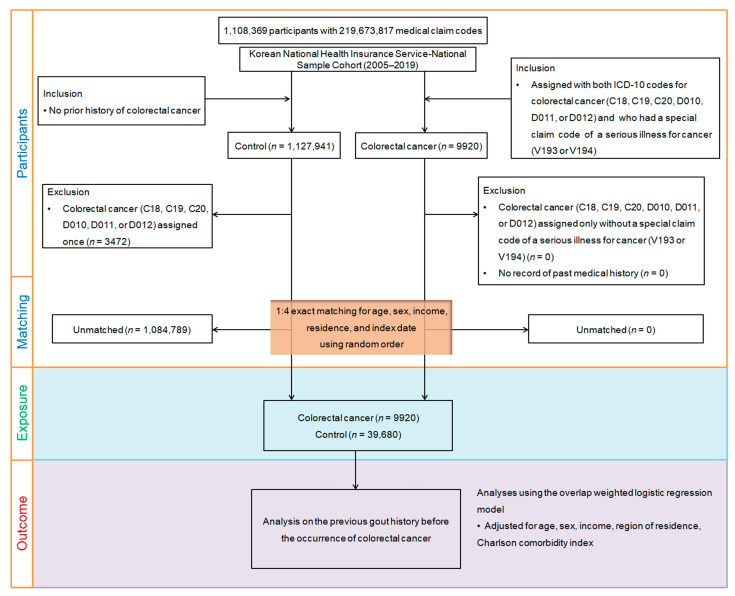
A visual representation of the participant selection procedure employed in this study. Out of the initial pool of 1,137,861 participants, 9920 individuals with colorectal cancer were carefully matched with 39,680 control participants based on factors such as age, sex, income, and residential region.

**Table 1 cancers-15-05602-t001:** General characteristics of participants before and after propensity score overlap weighting adjustment.

Characteristics		Before PS Overlap Weighting Adjustment			After PS Overlap Weighting Adjustment		
		CRC	Control	Standardized Difference	CRC	Control	Standardized Difference
Age (y), *n* (%)				0.00			0.00
	0–4	1 (0.01)	4 (0.01)		1 (0.01)	1 (0.01)	
	5–9	N/A	N/A		N/A	N/A	
	10–14	3 (0.03)	12 (0.03)		2 (0.03)	2 (0.03)	
	20–24	1 (0.01)	4 (0.01)		1 (0.01)	1 (0.01)	
	25–29	8 (0.08)	32 (0.08)		6 (0.08)	6 (0.08)	
	30–34	26 (0.26)	104 (0.26)		21 (0.26)	21 (0.26)	
	35–39	94 (0.95)	376 (0.95)		75 (0.94)	75 (0.94)	
	40–44	180 (1.81)	720 (1.81)		144 (1.81)	144 (1.81)	
	45–49	359 (3.62)	1436 (3.62)		286 (3.61)	286 (3.61)	
	50–54	578 (5.83)	2312 (5.83)		462 (5.82)	462 (5.82)	
	55–59	968 (9.76)	3872 (9.76)		773 (9.75)	773 (9.75)	
	60–64	1242 (12.52)	4968 (12.52)		992 (12.51)	992 (12.51)	
	65–69	1393 (14.04)	5572 (14.04)		1112 (14.03)	1112 (14.03)	
	70–74	1488 (15.00)	5952 (15.00)		1188 (15.00)	1188 (15.00)	
	75–79	1471 (14.83)	5884 (14.83)		1176 (14.84)	1176 (14.84)	
	80–84	1060 (10.69)	4240 (10.69)		849 (10.71)	849 (10.71)	
	85+	672 (6.77)	2688 (6.77)		538 (6.79)	538 (6.79)	
Sex, *n* (%)				0.00			0.00
	Male	5933 (59.81)	23,732 (59.81)		4739 (59.79)	4739 (59.79)	
	Female	3987 (40.19)	15,948 (40.19)		3186 (40.21)	3186 (40.21)	
Income, *n* (%)				0.00			0.00
	1 (lowest)	1990 (20.06)	7960 (20.06)		1589 (20.06)	1589 (20.06)	
	2	1253 (12.63)	5012 (12.63)		1000 (12.62)	1000 (12.62)	
	3	1562 (15.75)	6248 (15.75)		1247 (15.74)	1247 (15.74)	
	4	2059 (20.76)	8236 (20.76)		1646 (20.77)	1646 (20.77)	
	5 (highest)	3056 (30.81)	12,224 (30.81)		2442 (30.82)	2442 (30.82)	
Region of residence, *n* (%)				0.00			0.00
	Urban	4447 (44.83)	17,788 (44.83)		3553 (44.83)	3553 (44.83)	
	Rural	5473 (55.17)	21,892 (55.17)		4373 (55.17)	4373 (55.17)	
CCI score, mean (SD)		0.80 (1.18)	0.69 (1.18)	0.09	0.77 (1.03)	0.77 (0.57)	0.00
Gout, *n* (%)		339 (3.42)	1396 (3.52)	0.01	270 (3.40)	284 (3.58)	0.01

Abbreviations: PS, propensity score; CRC, colorectal cancer; N/A, not applicable; CCI, Charlson Comorbidity Index; SD, standard deviation.

**Table 2 cancers-15-05602-t002:** Crude and overlap propensity score-weighted odds ratios of gout for colorectal cancer and subgroup analyses according to age, sex, income, region of residence, and CCI scores.

Characteristics		N of CRC	N of Control	Odd Ratios for CRC (95% Confidence Interval)			
		(Exposure/Total, %)	(Exposure/Total, %)	Crude	*p*-Value	Overlap-Weighted Model †	*p*-Value
Total participants (*n* = 49,600)							
	Gout	339/9920 (3.4)	1396/39,680 (3.5)	0.97 (0.86–1.10)	0.628	0.95 (0.86–1.04)	0.282
	Control	9581/9920 (96.6)	38,284/39,680 (96.5)	1		1	
Age < 65 years (*n* = 24,265)							
	Gout	127/4853 (2.6)	590/19,412 (3.0)	0.86 (0.71–1.04)	0.12	0.82 (0.70–0.95)	0.009 *
	Control	4726/4853 (97.4)	18,822/19,412 (97.0)	1		1	
Age ≥ 65 years (*n* = 25,335)							
	Gout	212/5067 (4.2)	806/20,268 (4.0)	1.06 (0.90–1.23)	0.495	1.05 (0.93–1.19)	0.457
	Control	4855/5067 (95.8)	19,462/20,268 (96.0)	1		1	
Male (*n* = 29,665)							
	Gout	296/5933 (5.0)	1200/23,732 (5.1)	0.99 (0.87–1.12)	0.833	0.96 (0.87–1.07)	0.496
	Control	5637/5933 (95.0)	22,532/23,732 (94.9)	1		1	
Female (*n* = 19,935)							
	Gout	43/3987 (1.1)	196/15,948 (1.2)	0.88 (0.63–1.22)	0.435	0.85 (0.66–1.10)	0.223
	Control	3944/3987 (98.9)	15,752/15,948 (98.8)	1		1	
Low income group (*n* = 24,025)							
	Gout	169/4805 (3.5)	663/19,220 (3.4)	1.02 (0.86–1.21)	0.818	0.98 (0.86–1.13)	0.818
	Control	4636/4805 (96.5)	18,557/19,220 (96.6)	1		1	
High income group (*n* = 25,575)							
	Gout	170/5115 (3.3)	733/20,460 (3.6)	0.93 (0.78–1.10)	0.369	0.91 (0.80–1.05)	0.192
	Control	4945/5115 (96.7)	19,727/20,460 (96.4)	1		1	
Urban resident (*n* = 22,235)							
	Gout	140/4447 (3.1)	619/17,788 (3.5)	0.90 (0.75–1.09)	0.276	0.88 (0.76–1.02)	0.097
	Control	4307/4447 (96.9)	17,169/17,788 (96.5)	1		1	
Rural resident (*n* = 27,365)							
	Gout	199/5473 (3.6)	777/21,892 (3.5)	1.03 (0.88–1.20)	0.755	1.00 (0.88–1.14)	0.975
	Control	5274/5473 (96.4)	21,115/21,892 (96.5)	1		1	
CCI scores = 0 (*n* = 30,566)							
	Gout	164/5448 (3.0)	737/25,118 (2.9)	1.03 (0.86–1.22)	0.761	1.02 (0.90–1.17)	0.728
	Control	5284/5448 (97.0)	24,381/25,118 (97.1)	1		1	
CCI scores = 1 (*n* = 10,518)							
	Gout	74/2600 (2.8)	286/7918 (3.6)	0.78 (0.60–1.01)	0.063	0.81 (0.65–1.01)	0.065
	Control	2526/2600 (97.2)	7632/7918 (96.4)	1		1	
CCI scores ≥ 2 (*n* = 8516)							
	Gout	101/1872 (5.4)	373/6644 (5.6)	0.96 (0.77–1.20)	0.718	0.96 (0.79–1.15)	0.639
	Control	1771/1872 (94.6)	6271/6644 (94.4)	1		1	

Abbreviation: CRC, colorectal cancer; CCI, Charlson Comorbidity Index. * Significance at *p* < 0.05. † Adjusted for age, sex, income, region of residence, and CCI scores.

## Data Availability

Restrictions apply to the availability of these data. Data were obtained from the Korean National Health Insurance Sharing Service (NHISS) and are available at https://nhiss.nhis.or.kr (accessed on 25 January 2022) with the permission of the NHIS.

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
