# Peer review of "Gout and Colorectal Cancer Likelihood: Insights from a Nested Case-Control Study of the Korean Population Utilizing the Korean National Health Insurance Service-National Sample Cohort"

_cancers, 2023, doi:10.3390/cancers15235602_

Round 1

Reviewer 1 Report

Comments and Suggestions for Authors

Article Type: Research Article

Manuscript ID: cancers-2672665

Title: Gout and colorectal cancer likelihood: Insights from a nested case-control study of the Korean population utilizing the Korean National Health Insurance Service-National Sample Cohort

In this work, the authors studied whether Korean patient-specific factors such as age, gender, socioeconomic status, and presence of comorbidities may influence the relationship between gout and the likelihood of developing colorectal cancer using the Korean National Health Insurance Service-National Sample Cohort

I found that this manuscript is totally relevant for the field and brings interesting results that deserve consideration. However, certain points need to be revised before the manuscript could be considered for publication.

 -Why the number of patients (9920) and the number of witnesses (39,680) were not equally probable. This can influence the statistical study

 - It will be better if authors in the abstract add Background and aim, so that the reader better understand the goal of the study.

 - Are all the patients in this study of the same age?

- Does the study was carried out on the same stage of rectal cancer

-In retrospective studies, the samples must be varied from different regions, in this manuscript the choice of patients is not clear, and was it from different regions or from just one region? Please explain.

 -Examples of scans or echo graph that showed rectal cancer must be added to confirm this study.

- All p in the manuscript should be in italics throughout the manuscript

 - The conclusion should be summarized and rewritten

Comments on the Quality of English Language

Minor editing of English language required

Author Response

Reviewer #1:

General Comments: In this work, the authors studied whether Korean patient-specific factors such as age, gender, socioeconomic status, and presence of comorbidities may influence the relationship between gout and the likelihood of developing colorectal cancer using the Korean National Health Insurance Service-National Sample Cohort

I found that this manuscript is totally relevant for the field and brings interesting results that deserve consideration. However, certain points need to be revised before the manuscript could be considered for publication. 

Response: We extend our heartfelt gratitude to the reviewer for dedicating their time and expertise to thoroughly evaluate our manuscript. Your constructive feedback is greatly appreciated, and we value your insightful comments.

Comment 1: Why the number of patients (9920) and the number of witnesses (39,680) were not equally probable. This can influence the statistical study

Response: Conducting a case-control study for certain diseases can be challenging due to the need for a substantial number of study subjects, necessitating a diligent effort to increase the number of control subjects. Over the study period spanning from 2005 to 2019, we identified 9,920 CRC patients. To address baseline demographic and clinical disparities between the CRC and control groups, we employed propensity score matching. The matching process involved random selection of study subjects and pairing them with the control subject exhibiting the least disparity in propensity scores from among four control groups, commencing with the first subject in the CRC group. This 1:4 matching ratio was chosen for two primary reasons: first, 1:1 matching risked substantial data loss from the CRC group in cases where propensity scores were dissimilar between the control and CRC groups, and second, while 1:2 matching offers some advantages over 1:1 matching, research suggests that exceeding a 1:5 matching ratio provides minimal improvements in test power, with no appreciable gain in statistical power observed beyond a 1:4 control-to-case ratio in the study. (Biometrics. 1969 Jun;25(2):339-55.)

We added this description in the Materials and Methods, as like below.

(Materials and Methods, line 140): Given that research indicates minimal improvements in test power beyond a 1:5 matching ratio, and no appreciable gain in statistical power was observed beyond a 1:4 control-to-case ratio in the study, we opted for a 1:4 matching approach in our study.

Comment 2: It will be better if authors in the abstract add Background and aim, so that the reader better understand the goal of the study.

Response: We wholeheartedly agree with your opinion, but abstract should be a total of about 200 words maximum. Therefore, we changed it as follows.

(Abstract, line 35): Considering the global importance of both gout and colorectal cancer as significant health issues with mutual relevance, we aimed to examine the risk of colorectal cancer in Korean patients with gout.

Comment 3: Are all the patients in this study of the same age?

Response: As shown in Table 1, various age groups were included, and 1:4 propensity matching was conducted for the two groups in 5-year increments. We attempted to match an equal proportion of participants in each age group to reduce demographic heterogeneity.

Comment 4: Does the study was carried out on the same stage of rectal cancer. Examples of scans or echo graph that showed rectal cancer must be added to confirm this study.

Response: We could not separately carry out the analysis for the rectal cancers only. Conducting a comprehensive analysis based on colorectal cancer (CRC) staging and location is essential. Regrettably, our national cohort database did not include data pertaining to the staging and location of CRC, which represents a limitation of our study. Additionally, it is worth noting that conducting an in-depth analysis of CRC according to staging and location might potentially reduce the statistical power for verification. Therefore, should we gain access to a database encompassing a larger cohort of CRC patients in future studies, we will certainly take your valuable suggestion into account. We have included this limitation in the Discussion section, as follows:

(Discussion, line 349): Finally, the KNHIS-NSC database does not provide extensive information on factors such as the severity of gout, CRC stage, histologic characteristics, tumor progression, survival rates, family history, medication usage, dietary habits, and baseline and evolving uric acid levels during cancer development. Limitations include the absence of separate analyses for colon and rectal cancers with detailed staging and location data, as well as the lack of tumor stage and prognostic information. These limitations may hinder our ability to address potential confounding factors that were not measured, thereby limiting our analysis in accounting for these unmeasured variables. An additional limitation of our study is the inability to analyze body mass index and smoking status, which were not included in the analysis.

Comment 5: In retrospective studies, the samples must be varied from different regions, in this manuscript the choice of patients is not clear, and was it from different regions or from just one region? Please explain.

Response: We are sorry for our poor explanation. Initially, we collected information from various regions, and then, we categorized the regions of residence into two groups: urban, comprising the seven largest urban areas in Korea (Seoul, Incheon, Daejeon, Daegu, Gwangju, Ulsan, and Busan), each with a regional population exceeding one million, and rural, encompassing areas with regional populations of less than one million (Chungcheongbuk, Chungcheongnam, Gyeonggi, Gangwon, Gyeongsangbuk, Gyeongsangnam, Jeollabuk, Jeollanam, and Jeju). We added this explanation, as like below.

(Materials and Methods, line 169): Residential areas were classified into 16 counties according to administrative districts, with further classification into either urban or rural areas. Urban areas comprised the 7 largest cities in Korea (Seoul, Incheon, Daejeon, Daegu, Gwangju, Ulsan, and Busan), each with regional populations exceeding one million, while rural areas encompassed other regions with total populations below one million (Chungcheongbuk, Chungcheongnam, Gyeonggi, Gangwon, Gyeongsangbuk, Gyeongsangnam, Jeollabuk, Jeollanam, and Jeju).

Comment 6: All p in the manuscript should be in italics throughout the manuscript

Response: We appreciate your thorough review and feedback. As per your suggestion, we have modified the abstract to italicize "p=" for improved readability.

Comment 7: The conclusion should be summarized and rewritten.

Response: Thank you for your suggestion. We have revised our conclusion to incorporate additional information and results from our study, as follows:

(Conclusions, line 360): Our comprehensive study suggests that gout does not appear to significantly increase the risk of CRC in the overall Korean population. However, we observed variations in this risk association among specific demographic factors. Notably, within a subgroup of patients aged under 65 years, there was a moderately reduced probability of CRC development among those with gout. Conversely, no such associations were evident in subgroup analyses based on sex, income, residential location, or CCI scores, even among individuals aged 65 years or older. These findings highlight the importance of considering individual patient characteristics, particularly age, when evaluating CRC risk in patients with gout. While our study contributes valuable insights, further research is warranted to explore these associations in greater depth and explore potential mechanisms underlying these observations.

Reviewer 2 Report

Comments and Suggestions for Authors

The purpose of this manuscript is to investigate the impact of gout on colorectal cancer. However,to my knowledge,  many similar articles have been published. In addition, the content of this manuscript also needs to be enriched.

1.Do you compare the BMI between two groups of patients?

2.Subgroup analysis needs to be further enriched, such as BMI、smoking history.

3.The analysis of gout and CRC staging, tumor progression, and survival rate should be added to this manuscript.

Author Response

Reviewer #2:

General Comments: The purpose of this manuscript is to investigate the impact of gout on colorectal cancer. However,to my knowledge, many similar articles have been published. In addition, the content of this manuscript also needs to be enriched.

Response: We extend our heartfelt gratitude to the reviewer for engaging with our manuscript and providing valuable feedback. Your input is invaluable to us.

Comment 1: Do you compare the BMI between two groups of patients? Subgroup analysis needs to be further enriched, such as BMI, smoking history.

Response: Thank you for your valuable suggestion; we sincerely appreciate your input. While it would have been beneficial to investigate additional risk factors such as BMI and smoking history for colorectal cancer, our decision to prioritize the CCI score for appropriate propensity matching allowed us to balance the two study groups effectively. It is important to note that the sample cohort data from the National Health Insurance Service (NHIS) was not directly accessible to the authors. Access to this data was facilitated by either visiting the analysis center or through remote access for data analysis and result extraction. Unfortunately, we must acknowledge that the period of data accessibility has now concluded, and regrettably, we are unable to perform further analysis at this time due to this data expiration. We have duly included this limitation in the limitation section of our study, as follows:

(Discussion, line ): An additional limitation of our study is the inability to analyze body mass index and smoking status, which were not included in the analysis.

Comment 2: The analysis of gout and CRC staging, tumor progression, and survival rate should be added to this manuscript.

Response: As you said, a detailed analysis study based on CRC staging, tumor progression, and survival rate of CRC is needed, but unfortunately, data on staging and prognosis of CRC were not included in our national cohort database. We have duly included this limitation in the limitation section of our study, as follows:

(Discussion, line 349): Finally, the KNHIS-NSC database does not provide extensive information on factors such as the severity of gout, CRC stage, histologic characteristics, tumor progression, survival rates, family history, medication usage, dietary habits, and baseline and evolving uric acid levels during cancer development. Limitations include the absence of separate analyses for colon and rectal cancers with detailed staging and location data, as well as the lack of tumor stage and prognostic information. These limitations may hinder our ability to address potential confounding factors that were not measured, thereby limiting our analysis in accounting for these unmeasured variables. An additional limitation of our study is the inability to analyze body mass index and smoking status, which were not included in the analysis.

Reviewer 3 Report

Comments and Suggestions for Authors

This very interesting article, well designed, with a clear methodology, simple results, with an adequate discussion and consistent conclusions.

1.      What is the main question addressed by the research?

The main question of this research is to understand the potential effect of having gout on the risk of developing colorectal cancer.

2.      Do you consider the topic original or relevant in the field? Does it address a specific gap in the field?

There is evidence supporting the relationship between having gout and an increased risk of cancer, possibly due to the oxidative capacity of uric acid, which promotes chronic systemic inflammation that may play a significant role in the carcinogenesis process. There is extensive information on the link between persistent inflammation and various gastrointestinal cancers, including CRC. In this regard, there is a gap in the association between gout and CRC, as the results are contradictory. Many of these studies had unbalanced sample sizes and included cohorts with non-uniform data, which may have introduced selection bias.

3.      What does it add to the subject area compared with other published material?

This study is conducted in a large cohort, where different covariates are balanced through propensity score weighting. This minimizes study bias and makes it highly robust.

4.      What specific improvements should the authors consider regarding the methodology? What further controls should be considered?

The methodology is appropriate and robust. It does not require additional controls.

5.      Are the conclusions consistent with the evidence and arguments presented and do they address the main question posed?

The conclusion is consistent with the presented evidence, and although one might initially think that gout could be a risk factor, in the case of individuals under 65 years old, it has a slight protective effect, which is well explained by the authors with the information currently available. As the authors point out, this opens a new field of study, delving into the mechanisms that lead to this slight protective effect. This finding is a topic for further studies.

6.      Are the references appropriate?

The references are appropriate.

7.      Please include any additional comments on the tables and figures.

The tables are comprehensible. The format is a bit disorderly, but I believe this is because of the journal's requirements.

Author Response

Reviewer #3:

General Comments: This very interesting article, well designed, with a clear methodology, simple results, with an adequate discussion and consistent conclusions.

 Response: First of all, we extend our heartfelt gratitude to the reviewer for dedicating their time and expertise to evaluate our manuscript. We deeply value your constructive and insightful feedback, which has significantly contributed to the enhancement of our paper.

Comment 1: What is the main question addressed by the research?

Response: As gout is linked to various systemic comorbidities, including malignancies, the presence of these comorbidities can significantly impact the quality of life for individuals with gout. Therefore, preventing the development of gout-related comorbidities and mitigating their associated risks is essential for effective gout management. While an increasing body of research has suggested a potential connection between gout and the heightened risk of cancer, including colorectal cancer, the debate on this issue persists. Consequently, we sought to investigate the risk of colorectal cancer among the Korean population with gout, providing a more explicit rationale in the Introduction section, as follows.

(Introduction, line 97): In this study, our objective was to investigate the risk of CRC among Korean individuals with gout. We conducted a meticulously designed nested case-control study, utilizing comprehensive subgroup analyses based on data from the Korean national public healthcare system, in order to explore the potential impact of gout on the development of CRC.

Comment 2: Do you consider the topic original or relevant in the field? Does it address a specific gap in the field?

Response: Thank you for your comments. Our comprehensive study suggests that gout does not appear to significantly elevate the risk of CRC in the broader Korean population. Our findings align with two population-based studies conducted in Taiwan, utilizing data from the National Health Insurance database over different periods, both of which reported no significant association between gout and CRC risk.

However, our results contrast with previous research conducted in Sweden and Taiwan, where a 1.29-fold to 2.25-fold increased risk of CRC associated with gout was reported. Notably, the Swedish cohort study highlighted an increased CRC risk, particularly among male patients with gout. It's worth noting that these previous studies encompassed various cancer types and had limited sample sizes, without a specific focus on CRC. Additionally, they did not achieve precise balance between the study and control groups in terms of baseline sociodemographic and health characteristics. The inherent heterogeneity due to demographic disparities likely contributed to significant differences in the original research findings.

In our study, we employed nationwide population-based controls matched using propensity scores and adjusted using the overlap weighting method, meticulously balancing baseline characteristics to minimize potential study heterogeneity and selection bias. Furthermore, we accounted for potential confounding factors by conducting multivariate conditional logistic regression analysis. Our findings indicate that gout may not be associated with an increased risk of CRC, even after adjustments for confounding factors such as sex, age, residence, income status, and CCI score, aligning with the outcomes of previous studies. We described these explanations in the Discussion sections, as like below.

(Discussion, line 257): Our findings are in line with the results of two population-based studies in Taiwan that were based on the National Health Insurance database information during different periods and reported no significant association between gout and the risk of CRC.

(Discussion, line 271): Our findings are in contrast to those of previous studies conducted in Sweden and Taiwan that reported a 1.29-fold to 2.25-fold increased risk of CRC associated with gout. The Swedish cohort study highlighted an increased CRC risk, specifically for male patients with gout. These studies included a variety of cancer types with limited sample sizes and did not specifically focus on CRC. Previous studies did not achieve an exact balance between the study and control groups in terms of baseline sociodemographic and health characteristics. Heterogeneity due to demographic differences likely causes large differences in the original quality of the research group. We used nationwide population-based controls matched using propensity scores and adjusted using the overlap weighting method to accurately balance baseline characteristics, minimizing potential study heterogeneity and selection bias. Potential confounding factors were also adjusted by performing multivariate conditional logistic regression analysis. The present study revealed that gout may not be associated with an increased risk of CRC, even after adjusting for confounding factors, including sex, age, residence, income status, and CCI score, consistent with previous studies.

Comment 3: What does it add to the subject area compared with other published material?

Response: This study possesses several strengths that distinguish it from other published materials in the subject area. This study is conducted in a large cohort having a substantial sample size, comprising 9920 patients and 39,680 controls from a meticulously organized nationwide healthcare database, where different covariates are balanced through propensity score weighting, which effectively mitigated selection bias across key demographic factors, such as age, sex, income, residence, and CCI score among participants, thus closely replicating the characteristics typically found in randomized clinical trials. This minimizes study bias and makes it highly robust. Demographic heterogeneity within the participant pool may have influenced the nature of associations observed between the original characteristics of populations. We revised this explanation in the Discussion as like bellow.

(Discussion, line 335): This study benefits from a large cohort with a substantial sample size, involving 9920 patients and 39,680 controls, meticulously sourced from a nationwide healthcare database. The study employs sophisticated techniques, including propensity score weighting, to ensure a balance across various covariates. This approach effectively mitigates selection bias across critical demographic factors, including age, sex, income, residence, and CCI scores among participants. As a result, the study closely replicates the characteristics typically encountered in randomized clinical trials, significantly reducing the potential for study bias and enhancing its robustness.

Comment 4: What specific improvements should the authors consider regarding the methodology? What further controls should be considered?

Response: Conducting a case-control study for certain diseases can be challenging due to the need for a substantial number of study subjects, necessitating a diligent effort to increase the number of control subjects. Over the study period spanning from 2005 to 2019, we identified 9,920 CRC patients. To address baseline demographic and clinical disparities between the CRC and control groups, we employed propensity score matching. The matching process involved random selection of study subjects and pairing them with the control subject exhibiting the least disparity in propensity scores from among four control groups, commencing with the first subject in the CRC group. This 1:4 matching ratio was chosen for two primary reasons: first, 1:1 matching risked substantial data loss from the CRC group in cases where propensity scores were dissimilar between the control and CRC groups, and second, while 1:2 matching offers some advantages over 1:1 matching, research suggests that exceeding a 1:5 matching ratio provides minimal improvements in test power, with no appreciable gain in statistical power observed beyond a 1:4 control-to-case ratio in the study. (Reference: Biometrics. 1969 Jun;25(2):339-55.) We added this description in the Materials and Methods, as like below.

(Materials and Methods, line 140): Given that research indicates minimal improvements in test power beyond a 1:5 matching ratio, and no appreciable gain in statistical power was observed beyond a 1:4 control-to-case ratio in the study, we opted for a 1:4 matching approach in our study.

Comment 5: Are the conclusions consistent with the evidence and arguments presented and do they address the main question posed?

Response: We appreciate the reviewer's thoughtful consideration of our study's conclusions in relation to the presented evidence and arguments. We have taken this feedback seriously and have ensured that our revised conclusions align more closely with the evidence and address the main question posed in our study. We believe the revised conclusions provide a more accurate reflection of our findings and their implications.

(Conclusions, line 360): Our comprehensive study suggests that gout does not appear to significantly increase the risk of CRC in the overall Korean population. However, we observed variations in this risk association among specific demographic factors. Notably, within a subgroup of patients aged under 65 years, there was a moderately reduced probability of CRC development among those with gout. Conversely, no such associations were evident in subgroup analyses based on sex, income, residential location, or CCI scores, even among individuals aged 65 years or older. These findings highlight the importance of considering individual patient characteristics, particularly age, when evaluating CRC risk in patients with gout. While our study contributes valuable insights, further research is warranted to explore these associations in greater depth and explore potential mechanisms underlying these observations.

Comment 6: Are the references appropriate?

Response: Thank you for your comments. We thoroughly reviewed and reassessed all the references in our study to ensure their appropriateness and accuracy.

Comment 7: Please include any additional comments on the tables and figures. The tables are comprehensible. The format is a bit disorderly, but I believe this is because of the journal's requirements.

Response: We are grateful for the reviewer's valuable feedback regarding the tables and figures. In response to your advice, we have made the necessary modifications to the table. We sincerely apologize for any inconvenience caused by any errors in our work.

Figure 1. A visual representation of the participant selection procedure employed in this study. Out of the initial pool of 1,137,861 participants, 9920 individuals with colorectal cancer were carefully matched with 39,680 control participants based on factors such as age, sex, income, and residential region.

Table 1. General characteristics of participants before and after propensity score overlap weighting adjustment

Characteristics

Before PS overlap weighting adjustment

After PS overlap weighting adjustment

CRC

Control

Standardized

difference

CRC

Control

Standardized

difference

Age (y), n (%)

0.00

0.00

0–4

1 (0.01)

4 (0.01)

1 (0.01)

1 (0.01)

5–9

N/A

N/A

N/A

N/A

10–14

3 (0.03)

12 (0.03)

2 (0.03)

2 (0.03)

20–24

1 (0.01)

4 (0.01)

1 (0.01)

1 (0.01)

25–29

8 (0.08)

32 (0.08)

6 (0.08)

6 (0.08)

30–34

26 (0.26)

104 (0.26)

21 (0.26)

21 (0.26)

35–39

94 (0.95)

376 (0.95)

75 (0.94)

75 (0.94)

40–44

180 (1.81)

720 (1.81)

144 (1.81)

144 (1.81)

45–49

359 (3.62)

1436 (3.62)

286 (3.61)

286 (3.61)

50–54

578 (5.83)

2312 (5.83)

462 (5.82)

462 (5.82)

55–59

968 (9.76)

3872 (9.76)

773 (9.75)

773 (9.75)

60–64

1242 (12.52)

4968 (12.52)

992 (12.51)

992 (12.51)

65–69

1393 (14.04)

5572 (14.04)

1112 (14.03)

1112 (14.03)

70–74

1488 (15.00)

5952 (15.00)

1188 (15.00)

1188 (15.00)

75–79

1471 (14.83)

5884 (14.83)

1176 (14.84)

1176 (14.84)

80–84

1060 (10.69)

4240 (10.69)

849 (10.71)

849 (10.71)

85+

672 (6.77)

2688 (6.77)

538 (6.79)

538 (6.79)

Sex, n (%)

0.00

0.00

Male

5933 (59.81)

23,732 (59.81)

4739 (59.79)

4739 (59.79)

Female

3987 (40.19)

15,948 (40.19)

3186 (40.21)

3186 (40.21)

Income, n (%)

0.00

0.00

1 (lowest)

1990 (20.06)

7960 (20.06)

1589 (20.06)

1589 (20.06)

2

1253 (12.63)

5012 (12.63)

1000 (12.62)

1000 (12.62)

3

1562 (15.75)

6248 (15.75)

1247 (15.74)

1247 (15.74)

4

2059 (20.76)

8236 (20.76)

1646 (20.77)

1646 (20.77)

5 (highest)

3056 (30.81)

12,224 (30.81)

2442 (30.82)

2442 (30.82)

Region of residence, n (%)

0.00

0.00

Urban

4447 (44.83)

17,788 (44.83)

3553 (44.83)

3553 (44.83)

Rural

5473 (55.17)

21,892 (55.17)

4373 (55.17)

4373 (55.17)

CCI score, mean (SD)

0.80 (1.18)

0.69 (1.18)

0.09

0.77 (1.03)

0.77 (0.57)

0.00

Gout, n (%)

339 (3.42)

1396 (3.52)

0.01

270 (3.40)

284 (3.58)

0.01

Abbreviations: PS, propensity score; CRC, colorectal cancer; N/A, not applicable; CCI, Charlson Comorbidity Index; SD, standard deviation.

Table 2. Crude and overlap propensity score-weighted odds ratios of gout for colorectal cancer and subgroup analyses according to age, sex, income, region of residence, and CCI scores.

Characteristics

N of CRC

N of control

Odd ratios for CRC

(95% confidence interval)

(exposure/total, %)

(exposure/total, %)

Crude

p-value

Overlap-weighted

model †

p-value

Total participants (n = 49,600)

Gout

339/9920 (3.4)

1396/39,680 (3.5)

0.97 (0.86–1.10)

0.628

0.95 (0.86–1.04)

0.282

Control

9581/9920 (96.6)

38,284/39,680 (96.5)

1

1

Age < 65 years (n = 24,265)

Gout

127/4853 (2.6)

590/19,412 (3.0)

0.86 (0.71–1.04)

0.12

0.82 (0.70–0.95)

0.009*

Control

4726/4853 (97.4)

18,822/19,412 (97.0)

1

1

Age ≥ 65 years (n = 25,335)

Gout

212/5067 (4.2)

806/20,268 (4.0)

1.06 (0.90–1.23)

0.495

1.05 (0.93–1.19)

0.457

Control

4855/5067 (95.8)

19,462/20,268 (96.0)

1

1

Male (n = 29,665)

Gout

296/5933 (5.0)

1200/23,732 (5.1)

0.99 (0.87–1.12)

0.833

0.96 (0.87–1.07)

0.496

Control

5637/5933 (95.0)

22,532/23,732 (94.9)

1

1

Female (n = 19,935)

Gout

43/3987 (1.1)

196/15,948 (1.2)

0.88 (0.63–1.22)

0.435

0.85 (0.66–1.10)

0.223

Control

3944/3987 (98.9)

15,752/15,948 (98.8)

1

1

Low income group (n = 24,025)

Gout

169/4805 (3.5)

663/19,220 (3.4)

1.02 (0.86–1.21)

0.818

0.98 (0.86–1.13)

0.818

Control

4636/4805 (96.5)

18,557/19,220 (96.6)

1

1

High income group (n = 25,575)

Gout

170/5115 (3.3)

733/20,460 (3.6)

0.93 (0.78–1.10)

0.369

0.91 (0.80–1.05)

0.192

Control

4945/5115 (96.7)

19,727/20,460 (96.4)

1

1

Urban resident (n = 22,235)

Gout

140/4447 (3.1)

619/17,788 (3.5)

0.90 (0.75–1.09)

0.276

0.88 (0.76–1.02)

0.097

Control

4307/4447 (96.9)

17,169/17,788 (96.5)

1

1

Rural resident (n = 27,365)

Gout

199/5473 (3.6)

777/21,892 (3.5)

1.03 (0.88–1.20)

0.755

1.00 (0.88–1.14)

0.975

Control

5274/5473 (96.4)

21,115/21,892 (96.5)

1

1

CCI scores = 0 (n = 30,566)

Gout

164/5448 (3.0)

737/25,118 (2.9)

1.03 (0.86–1.22)

0.761

1.02 (0.90–1.17)

0.728

Control

5284/5448 (97.0)

24,381/25,118 (97.1)

1

1

CCI scores = 1 (n = 10,518)

Gout

74/2600 (2.8)

286/7918 (3.6)

0.78 (0.60–1.01)

0.063

0.81 (0.65–1.01)

0.065

Control

2526/2600 (97.2)

7632/7918 (96.4)

1

1

CCI scores ≥ 2 (n = 8516)

Gout

101/1872 (5.4)

373/6644 (5.6)

0.96 (0.77–1.20)

0.718

0.96 (0.79–1.15)

0.639

Control

1771/1872 (94.6)

6271/6644 (94.4)

1

1

Abbreviation: CRC, colorectal cancer; CCI, Charlson Comorbidity Index.

* Significance at p < 0.05

† Adjusted for age, sex, income, region of residence, and CCI scores.
